# Mitochondrial oxidative phosphorylation in peripheral blood mononuclear cells is decreased in chronic HIV and correlates with immune dysregulation

**Louie Mar A. Gangcuangco**[ORCID]**, Brooks I. Mitchell, Chathura Siriwardhana, Lindsay B. Kohorn, Glen M. Chew, Scott Bowler**[ORCID]**, Kalpana J. Kallianpur, Dominic C. Chow, Lishomwa C. Ndhlovu, Mariana Gerschenson, Cecilia M. Shikuma***

University of Hawaii, Honolulu, Hawaii, United States of America

* shikuma@hawaii.edu

## Abstract

### Background

Cellular immunometabolism among people living with HIV (PLWH) on antiretroviral therapy (ART) remains under investigated. We assessed the relationships between mitochondrial oxidative phosphorylation (OXPHOS) in peripheral blood mononuclear cells (PBMCs) and blood parameters associated with HIV immune dysregulation.

### Methods

PLWH $\geq$40 years old and on stable ART $\geq$3 months were enrolled (N = 149). OXPHOS complex I (CI, NADH dehydrogenase) and complex IV (CIV, cytochrome $c$ oxidase) protein levels in PBMCs were quantified using immunoassays. Monocyte subsets and markers of T-cell activation, senescence, and exhaustion were measured on PBMC by flow cytometry. Plasma inflammatory mediators were quantified using a multiplex assay. HIV-uninfected group (N = 44) of similar age, gender, and ethnicity had available OXPHOS levels.

### Results

PLWH had a median age of 51 years. Majority were male (88.6%), Caucasian (57.7%), and with undetectable plasma HIV RNA <50 copies/mL (84.6%). Median CI level was lower in PLWH compared with the HIV-seronegative group (65.5 vs 155.0 optical density/μg protein x $10^3$, p <0.0001). There was no significant difference in median CIV levels. Lower OXPHOS levels correlated with lower CD4% and CD4/CD8 ratio. On multivariable linear regression adjusted for age, current use of zidovudine/didanosine, and HIV RNA (detectable versus undetectable), lower OXPHOS levels were significantly associated with higher MPO, SAA, SAP, and sVCAM, and higher frequencies of intermediate (CD14$^{++}$CD16$^+$) monocytes and TIGIT+TIM3+ CD4 T-cell (p<0.01).

**Data Availability Statement:** All relevant data are within the paper and its Supporting Information files.

**Funding:** The study was made possible by the funding and clinical research support from National Institute of Health grants R01HL095135 (CMS), U54MD007584, U54MD007601 and P20GM113134 (MG). Details of the grants may be accessed through: https://report.nih.gov/index. aspx. The funders had no role in study design, data collection and analysis, decision to publish, or preparation of the manuscript.

**Competing interests:** The authors have declared that no competing interests exist.

## Conclusion

CI PBMC protein levels were decreased in PLWH on ART. Decreased OXPHOS correlated with disease severity and inflammation. Further studies on the relationship between immunometabolism and immune dysregulation in HIV are warranted.

## Background

Mitochondria are considered "powerhouses" of eukaryotic cells. They are ubiquitous organelles whose primary function is to produce energy in the form of adenosine triphosphate (ATP) through oxidative phosphorylation (OXPHOS). Enzyme complexes embedded in the inner mitochondrial membrane facilitate electron cascade that eventually generate ATP. These enzyme complexes include complex I (nicotinamide adenine dinucleotide [NADH]: ubiquinone oxidoreductase, CI), complex II (succinate dehydrogenase), complex III (ubiquinol-cytochrome c oxidoreductase or cytochrome $bc_1$), complex IV (cytochrome c oxidase, CIV), and complex V (ATP synthase) [1].

CI serves as the entry-point for majority of the electrons in the respiratory chain. As electrons cascade through CI to CIV, hydrogen ions are pumped into the intermembrane space, creating a proton-motive force across the inner mitochondrial membrane, which is then used by Complex V to generate ATP from adenosine diphosphate and inorganic phosphate. The components of the respiratory chain are multi-subunit complexes composed of up to 92 different structural proteins encoded by both maternally-derived mitochondrial DNA (mtDNA) and nuclear genes [2, 3].

Mutations in mitochondrial or nuclear DNA can impair cellular respiration. Cells and tissues with high-energy demand, such as the brain, nerves, retina, skeletal and cardiac muscle are especially vulnerable to defects in the electron transport chain [2]. Mutations in the components of the respiratory chain are associated with diseases such as Parkinson's and Huntington's disease, seizures, hypotonia, ophthalmoplegia, stroke-like episodes, muscle weakness, and cardiomyopathy [4]. Lower cellular respiration in peripheral blood mononuclear cells (PBMCs) was also reported among patients with chronic fatigue syndrome [5].

People living with HIV (PLWH) are at increased risk of mitochondrial dysfunction as a result of older mitochondrial-toxic antiretrovirals, direct viral toxicity, chronic inflammation, and concurrent comorbidities [6]. Older nucleoside reverse transcriptase inhibitors (NRTIs) are known to cause depletion of mitochondrial DNA via inhibition of the mitochondrial-specific DNA polymerase-γ. HIV replication has been associated with altered mtDNA transcription and reduced activity of mitochondrial respiratory complexes [7]. Protease inhibitors cause mitochondrial damage by increasing oxidative stress and reducing mitochondrial function [8].

The role of cellular immunometabolism in HIV remains under investigated. Utilizing banked specimens from a cohort of chronically HIV-infected adults on stable ART, we examined the relationships between cellular bioenergetics as determined by mitochondrial OXPHOS proteins in PBMCs and various plasma pro-inflammatory biomarkers, circulating monocyte subpopulations, and T-cell immune phenotypes.

## Materials and methods

### Participant recruitment

Mitochondrial OXPHOS parameters were assessed cross-sectionally from the Hawaii Aging with HIV Cardiovascular Disease cohort consisting of PLWH ≥40 years old, and on stable

ART for $\geq$3 months. Participants were recruited between the years 2009 and 2012. PLWH with active malignancy, acute infection, or AIDS-defining illness at the time of enrollment were excluded. A cohort of HIV-seronegative individuals were recruited as a comparator group. IRB approval was obtained from the University of Hawaii Human Studies Program. All participants provided written informed consent. All banked specimens and data collected from participants were anonymized and de-identified prior to analysis.

## Mitochondrial assessments

Quantitation immunoassays (Abcam, PLC, Cambridge, MA) were performed to quantify OXPHOS CI and CIV protein levels in viable PBMCs, as previously reported [9]. Cell viability was between 90–95%, determined using AOPI (acridine orange/propidium iodide). Each vial of viable PBMCs was thawed and washed in 0.5 ml of phosphate-buffered saline (PBS) twice before addition of 0.5 ml of ice-cold extraction buffer [1.5% lauryl maltoside, 25 mM HEPES (pH 7.4), 100 mM NaCl, plus protease inhibitors (Sigma, P-8340)]. Samples were mixed gently and kept on ice for 20 min, and then they were spun in a microcentrifuge at 16,400 rpm at 4˚C for 20 min to remove insoluble cell debris. The supernatant, an extract of detergent-solubilized cellular proteins, was then assayed with the OXPHOS immunoassays. All samples were loaded on the immunoassays with equal amounts of total cell protein using an amount previously established with control samples to generate signals within the linear range of the assay. Therefore, the resulting signal was directly proportional to the amount of OXPHOS protein or enzyme activity in the sample. Quantitation of the signal was done by densitometric scanning with a Hamamatsu ICA-1000 reader.

Mitochondrial DNA-specific 8-oxo-2'-deoxyguanosine (mt-specific 8-oxo-dG) was measured by the Gene Specific Repair Assay [10, 11], which identifies the frequency of oxidative changes in guanine nucleotides within the mitochondrial DNA–16 kb molecule. Quantitation of the break frequency of mt-specific 8-oxo-dG was based on comparing the undigested and digested mitochondrial DNA break frequencies calculated using the Poisson distribution, with units reported in break frequency (BF). Mt-specific 8-oxo-dG BF was available only among PLWH.

## Immune parameters in persons living with HIV

Multiparametric flow cytometry was performed on cryopreserved PBMCs to measure the frequencies of monocyte phenotypes [classical ($CD14^{++}CD16^{-}$), intermediate ($CD14^{++}CD16^{+}$), non-classical ($CD14^{low/+}CD16^{++}$)], and T-cells expressing activation (HLA-DR and CD38), exhaustion (PD-1, TIM-3 and TIGIT), and senescence ($CD57^{+}$) markers, as previously described [12, 13]. Data was acquired on a custom 4-laser BD LSR Fortessa Cell Analyzer and all compensation and gating analyses were performed in FlowJo analytical software. Plasma samples were assayed for target inflammatory markers, using high-sensitivity Milliplex assays (Human CVD panels, EMD Millipore, Billerica, MA). Plasma soluble inflammatory markers included C-reactive protein (CRP), interferon gamma (IFN-γ), interleukins (IL-1β, IL-6, IL-8, IL-10), monocyte chemotactic protein-1 (MCP-1), matrix metalloproteinase-9 (MMP-9), myeloperoxidase (MPO), serum amyloid A (SAA), serum amyloid P (SAP), soluble adhesion molecules (sE selectin, sICAM, sVCAM), tumor necrosis factor-alpha (TNF-α), tissue plasminogen activator inhibitor-1 (tPAI-1), and vascular endothelial growth factor (VEGF).

## Statistical analyses

Median CI and CIV protein levels were compared between HIV-positive patients and seronegative controls using independent samples Mann-Whitney U test. The correlation of various immune parameters with CI and CIV were assessed using Spearman's correlation (rho, *r*).

Immunologic parameters that were found to correlate with CI and CIV at the level of significance of $p < 0.01$ were further analysed using linear regression analyses. Immune parameters were log-transformed to improve normal distribution of values. Multiple linear regression models were adjusted for age, current use of zidovudine or didanosine, and undetectable plasma HIV RNA. Differences in the median levels of various immune parameters were compared among patients with detectable versus undetectable mt-specific 8-oxo-dG using independent samples Mann-Whitney U test. Statistical analyses were performed using the IBM SPSS statistics version 25.0 (Armonk, NY).

## Results

The demographic and immunologic parameters of the 149 PLWH are summarized in **Table 1**. The median age was 51 years. Majority were male, Caucasian, and had undetectable plasma HIV RNA <50 copies/mL. Median current CD4 count was 505 cells/μL. Compared with PLWH, the HIV-seronegative group (N = 44) had similar age (51.0 years in PLWH *vs* 54.2 years in HIV-seronegatives, p = 0.36), gender (88.6% *vs* 81.8% male, p = 0.25), and ethnicity (57.7% *vs* 61.4% Caucasian, p = 0.67).

Median CI protein level was significantly lower in PLWH compared with the HIV-seronegative group (65.5 vs 155.0 optical density/μg of protein x $10^3$, $p < 0.0001$; **Fig 1**). In contrast, there was no significant difference in median CIV protein levels (49.2 vs 46.0 optical density/μg of protein x $10^3$, p = 0.20).

Among PLWH, lower CI protein levels correlated with lower CD4 count ($r = 0.19$, p = 0.02), CD4% ($r = 0.18$, p = 0.02), and CD4/CD8 ratio ($r = 0.18$, p = 0.03) (**Table 2**). Similarly, lower CIV protein levels correlated with lower CD4% ($r = 0.18$, p = 0.02) and CD4/CD8 ratio ($r = 0.18$, p = 0.03).

On multivariable linear regression analyses, lower CD4% and lower CD4/CD8 ratio remained significantly associated with lower CI and CIV protein levels (**Table 3**). OXPHOS parameters did not correlate with absolute CD8 count, CD8%, or percentage of activated CD38+HLA-DR+ CD8 T-cell.

The most frequently used ART at the time of enrollment were: tenofovir disoproxil fumarate (72.5%), emtricitabine (65.6%), efavirenz (42.5%), ritonavir (39.4%), lamivudine (23.1%), and atazanavir (17.5%). Of the older NRTIs, 7.5% were on zidovudine, 1.9% were on didanosine, and none on stavudine or zalcitabine. No participant was on an integrase strand transfer inhibitor. Current didanosine use was associated with significantly lower median CI (23.5 vs 66.5 optical density/μg of protein, p = 0.03) and CIV (21.9 vs 49.5 optical density/μg of protein, p = 0.05) protein levels. There were no significant differences in mt-specific 8-oxo-dG BF by current ART use.

OXPHOS protein levels in PBMC correlated inversely with several markers of inflammation. Lower CI strongly correlated with higher IL-1β ($r = -0.36$, $p < 0.001$), MCP-1 ($r = -0.23$, p = 0.008), MPO ($r = -0.33$, $p < 0.001$), SAA ($r = -0.37$, $p < 0.001$), SAP ($r = -0.43$, $p < 0.001$), and sVCAM ($r = -0.27$, p = 0.002). Additionally, lower CI correlated with higher frequencies of intermediate monocytes ($r = -0.28$, p = 0.001) and dual-expressing TIGIT+TIM3+ CD4 T-cells ($r = -0.40$, p = 0.009). Similar correlations were found with PBMC CIV levels (**Table 2**).

On multivariable linear regression (**Table 3**), lower CI remained significantly associated with higher MCP-1, MPO, SAA, SAP, sVCAM, intermediate monocyte, and dual TIGIT +TIM3+ CD4 T-cell frequencies. CIV remained negatively associated with MPO, SAA, SAP, sVCAM, intermediate monocytes, and dual TIGIT+TIM3+ CD4 T-cell frequencies. Other NCR-bearing T-cells correlated with OXPHOS protein levels at $p < 0.05$ but did not reach the threshold of significance of $p < 0.01$ (**Table 2**). Further analyzing these NCR-bearing T-cells

**Table 1. Demographic and immunologic parameters of persons living with HIV (N = 149).**

| Demographic Parameters | |
|---|---|
| Age, years | 51.0 (46.0–57.5) |
| Male, n(%) | 132 (88.6%) |
| Caucasian ethnicity, n(%) | 86 (57.7%) |
| **Immunologic Parameters** | |
| Undetectable HIV RNA < 50 copies/ml, n(%) | 126 (84.6%) |
| Nadir CD4 T cell count, cells/ μL | 150 (39, 266) |
| CD4 T cell count, cells/μL | 505.0 (345.0–660.5) |
| CD4 T cell % | 29.0% (21.0% - 36.5%) |
| Activated CD8 T cell % (CD8$^+$ HLA-DR CD38$^+$) | 10.5% (7.6% - 17.0%) |
| **Negative checkpoint receptors (%)** | |
| PD1+ CD4 T cell | 41.1 (30.3–48.3) |
| TIGIT+ CD4 T cell | 22.6 (16.4–28.3) |
| TIM3+ CD4 T cell | 15.7 (11.9–19.2) |
| TIGIT+PD1+ CD4 T cell | 15.8 (11.0–20.9) |
| TIM3+PD1+ CD4 T cell | 6.49 (4.1–9.2) |
| TIGIT+TIM3+ CD4 T cell | 3.5 (2.6–5.2) |
| PD1+ CD8 T cell | 36.3 (25.7–47.3) |
| TIGIT+ CD8 T cell | 44 (36.4–59.4) |
| TIM3+ CD8 T cell | 21.9 (18.1–28.0) |
| TIGIT+PD1+ CD8 T cell | 24.4 (17.7–32.0) |
| TIM3+PD1+ CD8 T cell | 5.36 (3.7–7.7) |
| TIGIT+TIM3+ CD8 T cell | 8.64 (6.1–12.9) |
| **Senescent markers (%)** | |
| CD57+CD28- CD8 T cell | 17.3 (12.7–24.4) |
| CD57+CD28- CD4 T cell | 4.16 (1.1–9.9) |
| **Monocyte subsets (%)** | |
| Classical (CD14$^{++}$CD16$^-$) | 75.9 (70.4–81.9) |
| Intermediate (CD14$^{++}$CD16$^+$) | 1.6 (0.6–4.2) |
| Non-classical (CD14$^{low/+}$CD16$^{++}$) | 6.0 (4.0–9.1) |
| **Plasma soluble biomarkers** | |
| CRP, ng/mL | 10,886.3 (3,584.0–46,060.0) |
| IFN-γ, pg/mL | 0.7 (0.4–1.3) |
| IL-1β, pg/mL | 0.3 (0.3–0.3) |
| IL-6, pg/mL | 1.6 (0.9–2.5) |
| IL-8, pg/mL | 3.5 (2.7–4.4) |
| IL-10, pg/mL | 2.1 (0.8–5.0) |
| MCP-1, pg/mL | 147.0 (115.0–179.1) |
| MMP-9, ng/mL | 55.4 (37.3–87.0) |
| MPO, ng/mL | 16.4 (11.4–23.0) |
| SAA, ng/mL | 14,000 (3,933.2–46,224.0) |
| SAP, ng/mL | 88,660.3 (51,867.1– 181,582.0) |
| sE-selectin, ng/mL | 33.6 (23.1–48.0) |
| sICAM-1, ng/mL | 140.1 (110.0–172.0) |
| sVCAM-1, ng/mL | 1168.5 (931.0–1335.4) |
| TNF-α, pg/mL | 3.1 (1.8–4.5) |
| tPAI-1, ng/mL | 86.1 (68.0–113.4) |

(*Continued*)

**Table 1.** (Continued)

| VEGF, pg/mL | 24.4 (13.4–49.5) |
| --- | --- |
| **Mitochondrial assessments** | |
| Complex I, optical density/ug x $10^3$ | 65.5 (48.0–87.0) |
| Complex IV, optical density/ug x $10^3$ | 49.2 (37.3–64.3) |
| Presence of mitochondria-specific 8-Oxo-2'-deoxyguanosine (n, %) break frequency* | 85 (53.1%) |

Values shown are median (quartile 1, quartile 3), unless otherwise specified.

*Patients with available mt-specific 8-oxo-dG (N = 147)

using multivariable linear regression, lower CI was associated with higher frequencies of TIGIT+ CD4 T-cell (β = -0.35, p = 0.039) and PD1+TIGIT+ CD4 T-cell (β = -0.35, p = 0.037). Similarly, lower CIV was associated with higher frequencies of TIGIT+ CD4 T-cell (β = -0.38, p = 0.022), PD1+TIGIT+ CD4 T-cell (β = -0.41, p = 0.016), and PD1+TIM3+ CD4 T-cell (β = -0.37, p = 0.040).

The presence of mt-specific 8-oxo-dG BF was significantly associated with higher CI protein levels (69.48 vs 57.18 optical density/ug x $10^3$, p = 0.069), CIV protein levels (53.08 vs 43.33 optical density/ug x $10^3$, p = 0.004), and IL-6 (1.78 vs 1.17 pg/ml, p = 0.008), but with lower frequencies of dual TIGIT+TIM3+ CD8 T-cell (7.99% vs 12.90%, p = 0.019). No associations were found between mt-specific 8-oxo-dG BF and other immune parameters, such as CD4 percent, CD4/CD8 ratio, soluble plasma biomarkers, monocyte phenotypes, and NCR-bearing CD4 T cells, which correlated with CI and CIV.

## Discussion

In this cohort of older PLWH on stable ART, we found that mitochondrial CI protein levels were significantly decreased compared to HIV-seronegative persons. Lower CI and CIV protein levels in PBMCs correlated with disease severity among PLWH, as assessed by CD4 percent and CD4/CD8 ratio. Decreased PBMC CI and CIV protein levels were associated with higher plasma inflammatory markers and increased frequencies of intermediate monocytes and TIGIT+TIM3+ CD4 T-cells.

HIV proteins induce direct viral toxicity to the mitochondria. HIV-1 trans-activator (Tat) protein has been shown to cause rapid dissipation of the mitochondrial transmembrane potential and inactivates cytochrome *c* oxidase in mouse liver, heart, and brain [14]. Similarly, gp120 and Tat proteins induce mitochondrial fragmentation in neurons [15]. In our PLWH cohort, mitochondrial OXPHOS protein levels in PBMC, particularly CI, were significantly decreased compared with the seronegative group. This observation is likely due to several factors, such as direct mitochondrial toxicity of HIV, antiretrovirals, comorbidities, and persistent inflammation. Mitochondrial DNA, which is necessary for the synthesis of the components of the OXPHOS system, has also been reported to be significantly decreased in chronic HIV [16].

Persistent inflammation and immune dysregulation have been well documented among PLWH despite the use of suppressive ART [17]. We found that lower PBMC OXPHOS levels correlated with lower CD4/CD8 ratio, higher pro-inflammatory cytokine levels (MCP-1, MPO, SAA, SAP, and sVCAM), and higher percentages of the pro-inflammatory intermediate monocyte subset. These inflammatory mediators are known to induce mitochondrial dysfunction, which increases reactive oxygen species (ROS), leading to a vicious cycle of mitochondrial damage and inflammation [18].

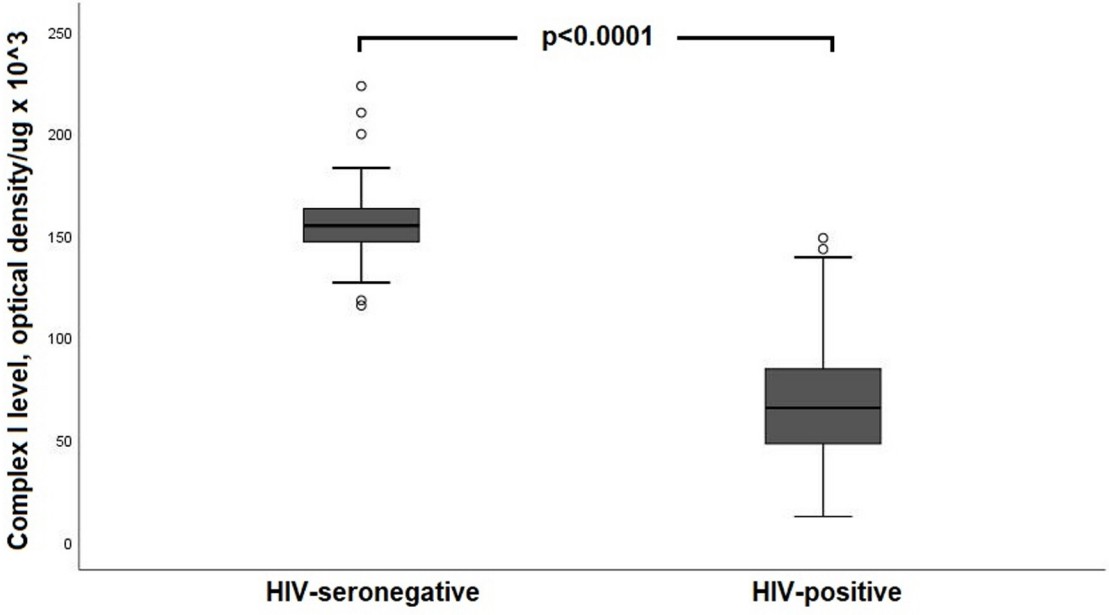

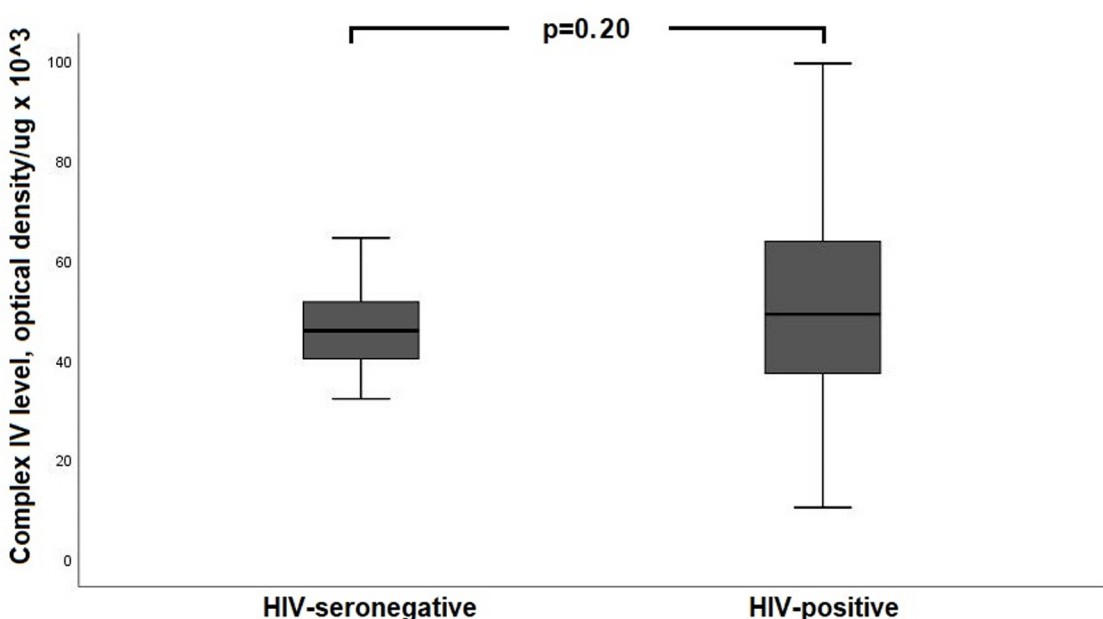

**Fig 1. Comparison of mitochondrial complex I and complex IV protein levels in peripheral blood mononuclear cells between HIV+ and seronegative group.**

Mitochondrial damage in chronic HIV is associated with increased lipid peroxidation of PBMC membranes [19]. Dysfunction of mitochondria can cause excess production of $O_2^-$ ions, resulting in a pro-inflammatory state [20]. Modulation of the inflammatory response via redox-sensitive pathways or direct activation of the inflammasome by mitochondria-derived free radicals may result in the production of cytokines and orchestrate a vigorous inflammatory response [21]. Mitochondrial dysfunction also increases cellular responsiveness to pro-

**Table 2. Spearman's correlation between mitochondrial oxidative phosphorylation protein levels in peripheral blood mononuclear cells and various immunologic parameters.**

| | Complex I (optical density (OD)/µg of protein$\times10^3$) | Complex IV (OD/µg of protein$\times10^3$) |
|---|---|---|
| Absolute CD4 count | 0.19 (p = 0.02) | 0.13 (p = 0.13) |
| CD4 percent | 0.18 (p = 0.02) | 0.18 (p = 0.02) |
| Absolute CD8 count | -0.06 (p = 0.50) | -0.10 (p = 0.25) |
| CD8 percent | -0.16 (p = 0.06) | -0.14 (p = 0.09) |
| CD4/CD8 ratio | 0.18 (p = 0.03) | 0.18 (p = 0.03) |
| CD38+HLA-DR+ CD8 T cell | -0.36 (p = 0.70) | 0.02 (p = 0.85) |
| **Plasma soluble biomarkers** | | |
| CRP | -0.01 (p = 0.93) | 0.15 (p = 0.20) |
| IFN-y | -0.08 (p = 0.37) | -0.09 (p = 0.33) |
| IL-1β | -0.36 (p<0.001) | -0.42 (p<0.001) |
| IL-6 | -0.11 (p = 0.22) | -0.05 (p = 0.56) |
| IL-8 | -0.07 (p = 0.43) | -0.04 (p = 0.65) |
| IL-10 | -0.02 (p = 0.81) | 0.01 (p = 0.89) |
| MCP-1 | -0.23 (p = 0.008) | -0.21 (p = 0.02) |
| MMP-9 | -0.18 (p = 0.03) | -0.11 (p = 0.23) |
| MPO | -0.33 (p<0.001) | -0.30 (p = 0.001) |
| SAA | -0.37 (p<0.001) | -0.37 (p<0.001) |
| SAP | -0.43 (p<0.001) | -0.47 (p<0.001) |
| sE-Selectin | -0.19 (p = 0.03) | -0.20 (p = 0.02) |
| sICAM | -0.12 (p = 0.18) | -0.10 (p = 0.24) |
| sVCAM | -0.27 (p = 0.002) | -0.32 (p<0.001) |
| TNF-a | -0.13 (p = 0.13) | -0.08 (p = 0.35) |
| tPAI-1 | -0.11 (p = 0.20) | -0.03 (p = 0.71) |
| VEGF | -0.04 (p = 0.61) | 0.02 (p = 0.77) |
| **Monocyte subsets (%)** | | |
| Classical (CD14++CD16−) | -0.19 (p = 0.03) | -0.15 (p = 0.08) |
| Intermediate (CD14++CD16+) | -0.28 (p = 0.001) | -0.27 (p = 0.002) |
| Non-classical (CD14low/+CD16++) | -0.09 (p = 0.32) | -0.15 (p = 0.08) |
| **Negative checkpoint receptors (%)** | | |
| PD1+ CD4 T-cell | -0.28 (p = 0.08) | -0.22 (p = 0.17) |
| TIGIT+ CD4+ T-cell | -0.33 (p = 0.03) | -0.35 (p = 0.02) |
| TIM3+ CD4 T-cell | -0.07 (p = 0.67) | -0.09 (p = 0.57) |
| PD1+ TIGIT+ CD4+ T-cell | -0.35 (p = 0.03) | -0.36 (p = 0.02) |
| PD1+ TIM3+ CD4 T-cell | -0.31 (p = 0.05) | -0.29 (p = 0.06) |
| TIGIT+ TIM3+ CD4 T-cell | -0.40 (p = 0.009) | -0.42 (p = 0.006) |
| TIGIT+ CD8 T-cell | -0.06 (p = 0.70) | -0.11 (p = 0.49) |
| PD1+ CD8 T-cell | -0.22 (p = 0.16) | -0.20 (p = 0.22) |
| TIM3+ CD8+ T-cell | -0.04 (p = 0.80) | -0.21 (p = 0.18) |
| PD1+ TIGIT+ CD8 T-cell | -0.14 (p = 0.39) | -0.16 (p = 0.31) |
| TIGIT+ TIM3+ CD8 T-cell | -0.16 (p = 0.31) | -0.32 (p = 0.04) |
| PD1+TIM3+ CD8 T-cell | -0.20 (p = 0.21) | -0.34 (p = 0.03) |
| **Senescence markers** | | |
| CD57+CD28- CD8 T cell | -0.15 (p = 0.33) | -0.17 (p = 0.29) |
| CD57+CD28- CD4 T cell | 0.16 (p = 0.32) | 0.34 (p = 0.03) |

inflammatory cytokines through an increase in ROS production [22] and NF-κB activation [23], and has been shown to cause accumulation of cells bearing pro-inflammatory phenotypes

**Table 3. Linear regression analyses of immunologic parameters associated with mitochondrial complex I and complex IV levels.**

| | Univariable | | Multivariable* | |
| --- | --- | --- | --- | --- |
| | Complex I | Complex IV | Complex I | Complex IV |
| Absolute CD4 count | 0.19 (p = 0.020) | 0.13 (p = 0.109) | 0.21 (p = 0.014) | 0.14 (p = 0.098) |
| CD4 percent | 0.20 (p = 0.017) | 0.18 (p = 0.028) | 0.21 (p = 0.015) | 0.19 (p = 0.026) |
| Absolute CD8 count | -0.05 (p = 0.578) | -0.08 (p = 0.318) | -0.03 (p = 0.700) | -0.08 (p = 0.342) |
| CD8 percent | -0.15 (p = 0.064) | -0.10 (p = 0.204) | -0.16 (p = 0.069) | -0.11 (p = 0.209) |
| CD4/CD8 ratio | 0.22 (p = 0.008) | 0.19 (p = 0.023) | 0.23 (p = 0.008) | 0.20 (p = 0.022) |
| IL-1β | 0.09 (p = 0.298) | -0.03 (p = 0.766) | 0.06 (p = 0.543) | -0.06 (p = 0.508) |
| MCP-1 | -0.21 (p = 0.017) | -0.15 (p = 0.087) | -0.24 (p = 0.008) | -0.17 (p = 0.059) |
| MPO | -0.24 (p = 0.006) | -0.30 (p = 0.001) | -0.25 (p = 0.005) | -0.31 (p<0.001) |
| SAA | -0.38 (p<0.001) | -0.34 (p<0.001) | -0.42 (p<0.001) | -0.37 (p<0.001) |
| SAP | -0.40 (p<0.001) | -0.44 (p<0.001) | -0.42 (p<0.001) | -0.45 (p<0.001) |
| sVCAM | -0.23 (p = 0.009) | -0.32 (p<0.001) | -0.25 (p = 0.006) | -0.33 (p<0.001) |
| Intermediate (CD14$^{++}$CD16$^{+}$) monocyte % | -0.25 (p = 0.003) | -0.23 (p = 0.007) | -0.26 (p = 0.002) | -0.23 (p = 0.007) |
| TIGIT+TIM3+ CD4 T-cell % | -0.42 (p = 0.006) | -0.49 (p = 0.001) | -0.42 (p = 0.014) | -0.52 (p = 0.002) |

Presented are β-coefficient and p-values.

*Separate multivariable linear regression analyses were performed for each immunologic parameter, adjusted for age, current use of didanosine/zidovudine, and undetectable HIV RNA. Plasma soluble inflammatory markers were log-transformed.

in lung tissue [24]. These inflammatory mediators could lead to further mitochondrial dysfunction [25]. The HIV proteins gp120, Tat, Nef, Vpr, and reverse transcriptase have been shown to enhance ROS production and dysregulate oxidative stress pathways [26].

CI levels in PLWH was decreased but CIV levels were similar to HIV-seronegative controls. The decreasing trend in dysfunction as the electrons progress through the respiratory chain has been previously reported. Among ART-naïve subjects, there was a 41% decrease in mitochondrial respiratory chain complex II activity, 38% decrease in complex III, and 19% decrease in CIV compared to seronegative controls [19].

We found that the presence of mitochondrial oxidative damage, as assessed by mt-specific 8-oxo-dG, was associated with *higher* CI and CIV protein levels. Interestingly, 8-oxo-dG, did not correlate with other immune parameters that correlated with CI and CIV, such as CD4 percent, monocyte phenotypes, or NCR-bearing CD4 T-cells. Dysfunctional OXPHOS leads to increased ROS generation, which damages mitochondrial DNA, membrane lipids, and proteins [27]. In particular, CI has been identified as a common site of superoxide generation [28, 29]. Our analyses were limited by the lack of mt-specific 8-oxo-dG levels in the HIV-seronegative group. In addition, we assessed OXPHOS protein levels and not function. One possible mechanism for our observation may be that increased ROS production is a consequence of stoichiometric mismatches in the electron transport chain complexes. This may result in increased residence time of electrons on sites of the complexes that mediate electron reduction of $O_2^-$ resulting in the increased production of $H_2O_2$ and superoxide [30]. We further hypothesize that the lack of changes in CIV protein levels in the setting of low CI may be due to the electron transport compensation via Complexes II-IV, which has been previously reported in computer modelling studies and other disease states [31, 32].

We found that higher percentage of TIM3+TIGIT+ CD4 T-cells was strongly associated with lower PBMC OXPHOS levels. TIM-3 and TIGIT are co-inhibitory receptors expressed by functionally exhausted T-cells, exhibiting decreased proliferation and suppressed T-cell responses [33, 34]. Aged T-cells upregulate the expression of co-inhibitory receptors and

demonstrate reduction in respiratory metabolism and electron transport chain activity [35]. We have previously reported that TIGIT expression on CD4 T-cells correlates with total HIV DNA and residual immune activation despite suppressive ART [13]. CD4 T-cells expressing TIGIT, as well as those expressing PD-1 or lymphocyte activation gene 3 (LAG-3), have been identified as a major contributor to the pool of inducible HIV genomes [36]. In the SPARTAC study, the expression on CD4 or CD8 T-cells of TIM-3 as well as expression of PD-1 or LAG-3 measured prior to ART predicted the time to viral rebound after treatment interruption [37]. PD-1 signaling has been shown to switch T cell metabolism from glycolysis to fatty acid metabolism. These metabolic reprogramming mediated by PD-1 may lead to mitochondrial depolarization, reduction of mitochondrial biogenesis, and higher rate of ROS production [38]. The effects of TIM-3 and TIGIT on the OXPHOS system warrant further investigation.

Several limitations of our analyses should be mentioned, including the cross-sectional nature of our study. Mt-specific 8-oxo-dG and other immune parameters (cytokines, monocyte subsets, and NCRs) were not available for the HIV-seronegative controls. In our cohort, CI and IV analyses were performed on PBMCs and not on individual cell populations. Although it has been described in the literature that T-cells constitute up to 90% of immune cells in PBMCs [39], the correlations between PBMC CI and CIV protein levels may not accurately reflect OXPHOS levels in T-cells. Furthermore, a recent review has described that various T-cell and monocyte subsets have unique metabolic profiles, with metabolically active effector CD4 T-cells predominantly utilizing glycolysis instead of OXPHOS [40]. Nonetheless, this study recruited a modest number of PLWH and identified key immune correlates of dysfunction important in inflammation and HIV pathogenesis.

In summary, we found that PBMC CI levels were decreased in PLWH on stable ART and correlated with HIV disease severity and inflammation. Decreased PBMC OXPHOS protein levels were associated with higher plasma inflammatory markers and increased frequencies of intermediate monocytes and TIGIT+TIM3+ CD4 T-cells. Further studies are needed to investigate the relationship between CD4 T-cell exhaustion, immunometabolism, and HIV persistence.

## Supporting information

**S1 File.**
(SAV)

## Acknowledgments

We thank the patients who participated in this study, as well as the staff of the Hawaii Center for AIDS for coordinating patient recruitment and gathering clinical data.

## Author Contributions

**Conceptualization:** Dominic C. Chow, Mariana Gerschenson, Cecilia M. Shikuma.

**Data curation:** Louie Mar A. Gangcuangco, Brooks I. Mitchell, Chathura Siriwardhana, Lindsay B. Kohorn, Glen M. Chew, Scott Bowler, Kalpana J. Kallianpur, Dominic C. Chow, Lishomwa C. Ndhlovu, Cecilia M. Shikuma.

**Formal analysis:** Louie Mar A. Gangcuangco, Brooks I. Mitchell, Chathura Siriwardhana, Lindsay B. Kohorn, Scott Bowler, Kalpana J. Kallianpur, Dominic C. Chow, Cecilia M. Shikuma.

**Funding acquisition:** Lishomwa C. Ndhlovu, Mariana Gerschenson, Cecilia M. Shikuma.

**Investigation:** Louie Mar A. Gangcuangco, Brooks I. Mitchell, Glen M. Chew, Dominic C. Chow, Lishomwa C. Ndhlovu, Mariana Gerschenson, Cecilia M. Shikuma.

**Methodology:** Louie Mar A. Gangcuangco, Brooks I. Mitchell, Chathura Siriwardhana, Glen M. Chew, Dominic C. Chow, Lishomwa C. Ndhlovu, Mariana Gerschenson, Cecilia M. Shikuma.

**Project administration:** Cecilia M. Shikuma.

**Resources:** Chathura Siriwardhana, Kalpana J. Kallianpur, Dominic C. Chow, Mariana Gerschenson, Cecilia M. Shikuma.

**Supervision:** Chathura Siriwardhana, Dominic C. Chow, Lishomwa C. Ndhlovu, Mariana Gerschenson, Cecilia M. Shikuma.

**Validation:** Louie Mar A. Gangcuangco, Chathura Siriwardhana, Lindsay B. Kohorn, Glen M. Chew, Dominic C. Chow, Cecilia M. Shikuma.

**Writing – original draft:** Louie Mar A. Gangcuangco, Brooks I. Mitchell, Lindsay B. Kohorn, Scott Bowler, Kalpana J. Kallianpur, Dominic C. Chow, Lishomwa C. Ndhlovu, Mariana Gerschenson, Cecilia M. Shikuma.

**Writing – review & editing:** Louie Mar A. Gangcuangco, Brooks I. Mitchell, Chathura Siriwardhana, Lindsay B. Kohorn, Glen M. Chew, Scott Bowler, Kalpana J. Kallianpur, Dominic C. Chow, Lishomwa C. Ndhlovu, Mariana Gerschenson, Cecilia M. Shikuma.

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
