## [Decision Letter · Decision Letter 0]

14 Nov 2019

PONE-D-19-24311

Mitochondrial oxidative phosphorylation in peripheral blood mononuclear cells is decreased in chronic HIV and correlates with immune dysregulation

PLOS ONE

Dear Dr. Louie Gangcuangco,

Thank you for submitting your manuscript to PLOS ONE. After careful consideration, we feel that it has merit but need a minor revision to fully meet PLOS ONE’s publication criteria. Therefore, we invite you to submit a revised version of the manuscript that addresses the points raised during the review process.

Reviewer #1

 Although the work is interesting and relevant to the overall knowledge in this field, I have some specific points which needs to be addressed.

1.     What is reason for not incorporating the HIV-seronegative data? Need to incorporate all the HIV-seronegative data (as PLHIV) in the table 1 for better comparison.

2.     How specific the method used for measuring Complex I? The authors need to incorporate more data using specific standards/inhibitors of both complex I and IV to measure their data set more stringently.

3.     Also, to make the case stronger the authors needs to incorporate immunostaining data (FACs) to measure both complex I & IV. What is the activity level of these enzymes?

4.     What is viability of the PBMC used?

5.     What is status of mitochondrial ROS in these samples as this is one of the important functional parameters?

6.     ‘We found that the presence of mitochondrial oxidative damage, as assessed by mt-specific 8-oxo-dG, was associated with higher CI and CIV protein levels’ – please explain elaborately as you didn’t find any changes on CIV.

7.     Needs more explanation in the discussion section about decreased complex I and unchanged complex IV. Reviewer #2This is an innovative study that examines a seldom explored facet of immunometabolism during HIV infection. The paper by Louie Mar A. Gangcuangco et al. explored the relationship between oxidative phosphorylation in peripheral blood mononuclear cells (PBMCs) and the frequency of monocyte subsets and markers of T-cell activation, senescence, and exhaustion. The study was made of 149 samples from people living with HIV (PLHIV) on ART and 44 samples from HIV-negative individuals. Oxidative Phosphorylation was quantified by measuring levels of Complex 1/4 proteins of the electron transport chain as well as an oxidative stress marker. The frequency of monocyte subsets and markers of T-cell activation, senescence, and exhaustion were analyzed using flow cytometry and levels of soluble inflammatory cytokines were determined using a multiplex assay. The PBMC levels of OXPHOS complex I were decreased in PLHIV on ART and the decreased OXPHOS correlated with disease severity and inflammation. This is an innovative study and the significance was well emphasized.

Some minor concerns are described as followed;

1) The study compared PLHIV undergoing ART+ and uninfected samples. There should be an additional control of samples from treatment-naive HIV infected individuals. Because ART can effect mitochondria during treatment, there needs to be control without ART to determine what influence ART had on the phenotype and correlation seen in the paper.

2) The paper uses levels of Complex 1/4 proteins as well as an oxidative stress marker as a measurement of mitochondrial dysregulation and OxPhos level. This is not an established method of determining either of the two. The study would be better served measuring the ratio of mitochondrial membrane polarization versus number of mitochondria to represent dysregulation. Moreover, functional analysis of mitochondrial metabolism (OxPhos) is most commonly done via Seahorse Oxygen Consumption Assays. Low levels of complex1/4 proteins may be mitigated by compensatory mechanisms. Directly measuring the levels of oxygen consumption as a marker for OxPhos will, therefore, be a much more accurate method than those used in the study.

We would appreciate receiving your revised manuscript by 02/14/2020. To enhance the reproducibility of your results, we recommend that if applicable you deposit your laboratory protocols in protocols.io, where a protocol can be assigned its own identifier (DOI) such that it can be cited independently in the future. For instructions see: http://journals.plos.org/plosone/s/submission-guidelines#loc-laboratory-protocols

We look forward to receiving your revised manuscript.

Kind regards,

Qigui Yu, M.D./Ph.D

Academic Editor

PLOS ONE

Journal Requirements:

Reviewers' comments:

Reviewer's Responses to Questions

**Comments to the Author**

1. Is the manuscript technically sound, and do the data support the conclusions?

Reviewer #1: Partly

Reviewer #2: Yes

2. Has the statistical analysis been performed appropriately and rigorously? 

Reviewer #1: Yes

Reviewer #2: Yes

3. Have the authors made all data underlying the findings in their manuscript fully available?

Reviewer #1: Yes

Reviewer #2: Yes

4. Is the manuscript presented in an intelligible fashion and written in standard English?

Reviewer #1: Yes

Reviewer #2: Yes

5. Review Comments to the Author

Reviewer #1: This is an innovative study that examines a seldom explored facet of immunometabolism during HIV infection. The paper by Louie Mar A. Gangcuangco et al. explored the relationship between oxidative phosphorylation in peripheral blood mononuclear cells (PBMCs) and the frequency of monocyte subsets and markers of T-cell activation, senescence, and exhaustion. The study was made of 149 samples from people living with HIV (PLHIV) on ART and 44 samples from HIV-negative individuals. Oxidative Phosphorylation was quantified by measuring levels of Complex 1/4 proteins of the electron transport chain as well as an oxidative stress marker. The frequency of monocyte subsets and markers of T-cell activation, senescence, and exhaustion were analyzed using flow cytometry and levels of soluble inflammatory cytokines were determined using a multiplex assay. The PBMC levels of OXPHOS complex I were decreased in PLHIV on ART and the decreased OXPHOS correlated with disease severity and inflammation. This is an innovative study and the significance was well emphasized.

Some minor concerns are described as followed;

1) The study compared PLHIV undergoing ART+ and uninfected samples. There should be an additional control of samples from treatment-naive HIV infected individuals. Because ART can effect mitochondria during treatment, there needs to be control without ART to determine what influence ART had on the phenotype and correlation seen in the paper.

2) The paper uses levels of Complex 1/4 proteins as well as an oxidative stress marker as a measurement of mitochondrial dysregulation and OxPhos level. This is not an established method of determining either of the two. The study would be better served measuring the ratio of mitochondrial membrane polarization versus number of mitochondria to represent dysregulation. Moreover, functional analysis of mitochondrial metabolism (OxPhos) is most commonly done via Seahorse Oxygen Consumption Assays. Low levels of complex1/4 proteins may be mitigated by compensatory mechanisms. Directly measuring the levels of oxygen consumption as a marker for OxPhos will, therefore, be a much more accurate method than those used in the study.

Reviewer #2: Although the work is interesting and relevant to the overall knowledge in this field, I have some specific points which needs to be addressed.

1. What is reason for not incorporating the HIV-seronegative data? Need to incorporate all the HIV-seronegative data (as PLHIV) in the table 1 for better comparison.

2. How specific the method used for measuring Complex I? The authors need to incorporate more data using specific standards/inhibitors of both complex I and IV to measure their data set more stringently.

3. Also, to make the case stronger the authors needs to incorporate immunostaining data (FACs) to measure both complex I & IV. What is the activity level of these enzymes?

4. What is viability of the PBMC used?

5. What is status of mitochondrial ROS in these samples as this is one of the important functional parameters?

6. ‘We found that the presence of mitochondrial oxidative damage, as assessed by mt-specific 8-oxo-dG, was associated with higher CI and CIV protein levels’ – please explain elaborately as you didn’t find any changes on CIV.

7. Needs more explanation in the discussion section about decreased complex I and unchanged complex IV.

6. PLOS authors have the option to publish the peer review history of their article (what does this mean?). If published, this will include your full peer review and any attached files.

Reviewer #1: Yes: Fahim Syed

Reviewer #2: No

---

## [Author Response · Author response to Decision Letter 0]

14 Feb 2020

February 9, 2020

Qigui Yu, MD, PhD

Academic Editor

PLOS ONE 

Re: Mitochondrial oxidative phosphorylation in peripheral blood mononuclear cells is decreased in chronic HIV and correlates with immune dysregulation (PONE-D-19-24311)

Dear Dr. Yu:

We would like to thank you and the Reviewers for their careful review of our manuscript. The reviewers thought the work was ‘interesting and relevant to the overall knowledge in the field’ and ‘innovative and significant’. We have addressed their minor revisions below and in the revised manuscript. 

Reviewer 1

Although the work is interesting and relevant to the overall knowledge in this field, I have some specific points which needs to be addressed.

1. What is reason for not incorporating the HIV-seronegative data? Need to incorporate all the HIV-seronegative data (as PLHIV) in the table 1 for better comparison.

Unfortunately, mt-specific 8-oxo-dG and other immune parameters (cytokines, monocyte subsets, and NCRs) were not available for the HIV-seronegative controls. We included this in the limitations section in the Discussion. Table 1 would therefore have several missing cells if an HIV-seronegative column is included. To compare and clarify the demographic characteristics of HIV-seronegatives, the following has been added to the first paragraph of the Results section (lines 158-162):

“Compared with PLHIV, the HIV-seronegative group (N=44) had similar age (51.0 years in PLHIV vs 54.2 years in HIV-seronegatives, p=0.36), gender (88.6% vs 81.8% male, p=0.25), and ethnicity (57.7% vs 61.4% Caucasian, p=0.67).”

2. How specific the method used for measuring Complex I? The authors need to incorporate more data using specific standards/inhibitors of both complex I and IV to measure their data set more stringently.

Complex I was measured as described in reference 9: PMCID: PMC2649940. Complex I and IV protein levels were measured and not activity, thus inhibitors were not used (e.g. rotenone for Complex I activity, etc.). Additional text describing the methodology has been added to lines 112-122.

3. Also, to make the case stronger the authors need to incorporate immunostaining data (FACs) to measure both complex I & IV. What is the activity level of these enzymes?

We agree with the reviewer that immunostaining of the PBMCS for Complex I and IV would have been feasible. We did not have enough PBMCs for both FACS and the immunoassays. We measured protein levels, not activity. 

4. What is viability of the PBMC used? 

The viability of the cells was 90-95%. Cell viability was determined using AOPI (acridine orange/propidium iodide). A statement was included in the Methods section. 

5. What is status of mitochondrial ROS in these samples as this is one of the important functional parameters?

We measured mitochondrial ROS by measuring the amount of oxidized guanine in the mtDNA. We added a reference in the Methods section for PMID 19321503, Gerschenson et al. 

Mitochondrial 8-oxo-dG damage was assessed using a Gene-Specific Repair Assay. Ten micrograms of PBMC DNA was isolated with a DNeasy Blood and Tissue Kit (Qiagen, Inc.). DNA was then digested with PvuII (New England BioLabs, Inc., Ipswich, MA, USA) overnight to linearize mtDNA. Digested DNA was halved: 5 µg of DNA was treated with human 8-oxoguanine DNA glycosylase (hOGG1) for 1 h at 37°C in a reaction volume of 15 µL and then for 1 h at 65°C for enzyme deactivation, and the remaining 5 µg of DNA was left untreated and stored at 4°C. For analysis, 4 µL of 1× Alkaline Agarose Loading Dye (Boston Bioproducts, MA, USA) was added, and cleaved and non-cleaved products were resolved on a 0.75% alkaline agarose gel. DNA was transferred to nylon (+) membranes using standard Southern blot methodology. Human mitochondrial probes specific for cytochrome b were labelled with digoxigenin-dUTP (Roche) by linear PCR amplification. Primer sequences were: DigFor, GCT ACC TTC ACG CCA A (14 976–15 001); and DigRev, CCG TTT CGT GCA AGA AT (15 357–15 341). Blots were hybridized overnight at 45°C and processed for chemiluminescent detection following Roche protocols. Finally, membranes were developed on a chemilumi imager (Roche) using LumiAnalyst software (Roche). Mitochondrial 8-oxo-dG damage was quantified by calculating break frequencies (BFs) based on the Poisson distribution of DNA treated with the hOGG1 repair enzyme and DNA untreated.

6. ‘We found that the presence of mitochondrial oxidative damage, as assessed by mt-specific 8-oxo-dG, was associated with higher CI and CIV protein levels’ – please explain elaborately as you didn’t find any changes on CIV.

We hypothesize that the lack of changes in Complex IV protein levels may be due to the electron transport compensation via Complexes II-IV. Text was added in Lines 272 to 275. 

7. Needs more explanation in the discussion section about decreased complex I and unchanged complex IV.

See lines 271 and 272. For instance, decreased Complex I activity in skeletal muscle has been associated with normalized Complex IV in human peripheral arterial disease (Pubmed: 11158957).

Reviewer #2

This is an innovative study that examines a seldom explored facet of immunometabolism during HIV infection. The paper by Louie Mar A. Gangcuangco et al. explored the relationship between oxidative phosphorylation in peripheral blood mononuclear cells (PBMCs) and the frequency of monocyte subsets and markers of T-cell activation, senescence, and exhaustion. The study was made of 149 samples from people living with HIV (PLHIV) on ART and 44 samples from HIV-negative individuals. Oxidative Phosphorylation was quantified by measuring levels of Complex 1/4 proteins of the electron transport chain as well as an oxidative stress marker. The frequency of monocyte subsets and markers of T-cell activation, senescence, and exhaustion were analyzed using flow cytometry and levels of soluble inflammatory cytokines were determined using a multiplex assay. The PBMC levels of OXPHOS complex I were decreased in PLHIV on ART and the decreased OXPHOS correlated with disease severity and inflammation. This is an innovative study and the significance was well emphasized.

Some minor concerns are described as followed;

1) The study compared PLHIV undergoing ART+ and uninfected samples. There should be an additional control of samples from treatment-naive HIV infected individuals. Because ART can effect mitochondria during treatment, there needs to be control without ART to determine what influence ART had on the phenotype and correlation seen in the paper.

We concur with the reviewer that it would be useful to have a treatment-naive HIV infected individuals assayed. These PBMCs are unavailable in our current banked specimens, since most PLHIV are treated as soon as they are diagnosed per CDC recommendations (https://www.helpstoptheviruspro.com/start-hiv-treatment/treat-hiv-asap?sitelink=treating+hiv+asap&utm_medium=cpc&utm_campaign=2019_Treatment+-+Broad&utm_content=Strategy&utm_term=%2Bguideline+%2Bhiv&moc=HSTV01&utm_source=bing&gclid=CK_F48GtxecCFTuJxQIdNHcCVg&gclsrc=ds). 

2) The paper uses levels of Complex 1/4 proteins as well as an oxidative stress marker as a measurement of mitochondrial dysregulation and OxPhos level. This is not an established method of determining either of the two. The study would be better served measuring the ratio of mitochondrial membrane polarization versus number of mitochondria to represent dysregulation. Moreover, functional analysis of mitochondrial metabolism (OxPhos) is most commonly done via Seahorse Oxygen Consumption Assays. Low levels of complex1/4 proteins may be mitigated by compensatory mechanisms. Directly measuring the levels of oxygen consumption as a marker for OxPhos will, therefore, be a much more accurate method than those used in the study.

We agree with the Reviewer that that the Seahorse assays and membrane polarization would have given mechanistic information. We developed the OXPHOS immunoassays (reference 9, 26762648, 26002840) and have used the mitochondrial 8-oxo-guanine assays (PMID 19321503, 2257461, 25574857) as an initial assay to evaluate mitochondrial function. We do not have viable PBMCs to conduct Seahorse experiments and will do so in future studies. 

Once again, we thank the reviewers for their valuable feedback. We hope that we have addressed the queries adequately. 

Sincerely,

Louie Mar Gangcuangco, MD

---

## [Decision Letter · Decision Letter 1]

1 Apr 2020

Mitochondrial oxidative phosphorylation in peripheral blood mononuclear cells is decreased in chronic HIV and correlates with immune dysregulation

PONE-D-19-24311R1

Dear Dr. Gangcuangco,

We are pleased to inform you that your manuscript has been judged scientifically suitable for publication and will be formally accepted for publication once it complies with all outstanding technical requirements.

With kind regards,

Qigui Yu, M.D./Ph.D

Academic Editor

PLOS ONE

Additional Editor Comments (optional):

Reviewers' comments:

Reviewer's Responses to Questions

**Comments to the Author**

1. If the authors have adequately addressed your comments raised in a previous round of review and you feel that this manuscript is now acceptable for publication, you may indicate that here to bypass the “Comments to the Author” section, enter your conflict of interest statement in the “Confidential to Editor” section, and submit your "Accept" recommendation.

Reviewer #1: All comments have been addressed

Reviewer #2: All comments have been addressed

2. Is the manuscript technically sound, and do the data support the conclusions?

Reviewer #1: Yes

Reviewer #2: Yes

3. Has the statistical analysis been performed appropriately and rigorously? 

Reviewer #1: Yes

Reviewer #2: Yes

4. Have the authors made all data underlying the findings in their manuscript fully available?

Reviewer #1: Yes

Reviewer #2: Yes

5. Is the manuscript presented in an intelligible fashion and written in standard English?

Reviewer #1: Yes

Reviewer #2: Yes

6. Review Comments to the Author

Reviewer #1: (No Response)

Reviewer #2: (No Response)

7. PLOS authors have the option to publish the peer review history of their article (what does this mean?). If published, this will include your full peer review and any attached files.

Reviewer #1: Yes: Fahim Syed

Reviewer #2: No

---

## [Editor Report · Acceptance letter]

20 Apr 2020

PONE-D-19-24311R1 

Mitochondrial oxidative phosphorylation in peripheral blood mononuclear cells is decreased in chronic HIV and correlates with immune dysregulation 

Dear Dr. Gangcuangco:

I am pleased to inform you that your manuscript has been deemed suitable for publication in PLOS ONE. Congratulations! Your manuscript is now with our production department. 

With kind regards,

on behalf of

Dr. Qigui Yu 

Academic Editor

PLOS ONE